# Fluorescence fluctuation analysis reveals *PpV* dependent Cdc25 protein dynamics in living embryos

**Boyang Liu**[1,2]*, **Ingo Gregor**[3], **H.-Arno Müller**[4], **Jörg Großhans**[1,2]*

**1** Fachbereich Biologie (FB17), Philipps-Universität Marburg, Marburg, Germany, **2** Institut für Entwicklungsbiochemie, Universitätsmedizin, Georg-August-Universität Göttingen, Göttingen, Germany, **3** Drittes Physikalisches Institut, Georg-August-Universität Göttingen, Göttingen, Germany, **4** Fachgebiet Entwicklungsgenetik, Institut für Biologie, Universität Kassel, Kassel, Germany

\* grosshan@uni-marburg.de (JG); boyang.liu@med.uni-goettingen.de (BL)

## Abstract

The protein phosphatase Cdc25 is a key regulator of the cell cycle by activating Cdk-cyclin complexes. Cdc25 is regulated by its expression levels and post-translational mechanisms. In early *Drosophila* embryogenesis, Cdc25/Twine drives the fast and synchronous nuclear cycles. A pause in the cell cycle and the remodeling to a more generic cell cycle mode with a gap phase are determined by Twine inactivation and destruction in early interphase 14, in response to zygotic genome activation. Although the pseudokinase Tribbles contributes to the timely degradation of Twine, Twine levels are controlled by additional yet unknown post-translational mechanisms. Here, we apply a non-invasive method based on fluorescence fluctuation analysis (FFA) to record the absolute concentration profiles of Twine with min-ute-scale resolution in single living embryos. Employing this assay, we found that Protein phosphatase V (PpV), the homologue of the catalytic subunit of human PP6, ensures appro-priately low Twine protein levels at the onset of interphase 14. *PpV* controls directly or indi-rectly the phosphorylation of Twine at multiple serine and threonine residues as revealed by phosphosite mapping. Mutational analysis confirmed that these sites are involved in control of Twine protein dynamics, and cell cycle remodeling is delayed in a fraction of the phospho-site mutant embryos. Our data reveal a novel mechanism for control of Twine protein levels and their significance for embryonic cell cycle remodeling.

## Author summary

Embryonic development starts with a series of fast nuclear divisions in most animals, which is followed by a dramatical cell cycle slowdown to enter a pause. *Drosophila* embryos undergo 13 fast and synchronous nuclear cycles with only S and M phases. In interphase 14, the cell cycle is remodeled: the mitosis pauses, the S phase is prolonged, and a gap phase is introduced. Post-translational regulation of Cdc25/Twine phosphatase is responsible for this remodeling. Although *tribbles* is involved, it has remained unclear how the timely degradation of Twine in interphase 14 is controlled. Here, we show that

**Data Availability Statement:** All relevant data are within the manuscript and its Supporting Information files.

**Funding:** This work was in part supported by the Deutsche Forschungsgemeinschaft (DFG GR1945/

3-1, GR1945/14-1, and equipment grant INST1525/16-1 FUGG). Work in the HAM laboratory was supported by a project grant (MRC K018531/1) from the Medical Research Council (U. K.). BL was in part supported by a fellowship from the China Scholarship Council. The funders had no role in study design, data collection and analysis, decision to publish, or preparation of the manuscript.

**Competing interests:** The authors have declared that no competing interests exist.

*Protein phosphatase V* (*PpV*) tightly controls Twine dynamics and thus the timing of cell cycle remodeling. *PpV* ensures appropriately low Twine levels at the onset of interphase 14 with little embryo-to-embryo variation. During interphase 14, *tribbles* and other factors are involved in the swift Twine degradation. This study provides insights in post-translational and safeguarding mechanisms of Cdc25, as well as in developmental timing of early embryogenesis.

## Introduction

The progression of cell cycle and the change between cell cycle modes are controlled by activity and expression levels of regulatory proteins. The antagonistic pair of Wee1/Myt1 kinases and Cdc25 phosphatase is a paradigm for the post-translational control of the Cdk1-cyclin complex, acting on the T14Y15 sites of Cdk1 [1, 2]. Wee1 and Cdc25 themselves are regulated by phosphorylation and protein levels [3, 4]. Besides during normal cell cycle progression, the control of their activity and protein levels is especially important during remodeling of the cell cycle, such as during developmental transitions or in coordination with morphogenesis. A prominent and well-studied example is the remodeling during the transition from syncytial to cellular development in early *Drosophila* embryos, when the rapid nuclear cycle changes to a more generic mode with a G2 phase.

The 13 rounds of nuclear cycle proceed with fast S phases but lack gap phases and cytokinesis. After mitosis 13, the cell cycle changes to a mode with cytokinesis, a slow S phase, and a G2 phase [5, 6]. This change from fast nuclear cycles to slow embryonic cell cycle is linked to the onset of zygotic gene transcription, the degradation of maternal RNAs, consumption of maternal metabolites, and morphological changes [7–11]. The onset of zygotic transcription is necessary and sufficient for remodeling the cell cycle in that the DNA replication checkpoint is activated by an interference of transcription and replication [8, 12]. Furthermore, specific zygotic genes encoding mitotic inhibitors contribute to the inactivation of Cdk1-cyclin and cell cycle remodeling [13–15].

The post-translational control of Cdc25 is at the center of cell cycle remodeling regulation [16–19] (Fig 1A). *Drosophila* Cdc25 homologues are encoded by two genes, *string* and *twine*. Although both are abundantly expressed in mature oocytes and during nuclear cycles, Twine is specifically required for the nuclear cycles [18, 19]. Timely degradation of Twine in response to zygotic genome activation correlates to the decline of Cdk1 activity and consequent cell cycle remodeling [19–21]. The mechanism for the timely decay of Twine is unknown, yet the pseudokinase Tribbles (Trbl) makes a contribution [14, 22, 23]. *Trbl* antagonizes Cdc25/String and promotes Cdc25/Twine degradation in interphase 14 [15, 19]. However, the molecular mechanism of *trbl* function is poorly understood.

In addition to *trbl*, Twine may be post-translationally controlled by other factors, since cell cycle remodeling does not exclusively depend on *trbl* (Fig 1A). In other systems, Cdc25 activity can be modulated through its phosphorylation state by other kinases and phosphatases at multiple amino acid residues [24]. *Protein phosphatase V* (*PpV*) encodes the *Drosophila* homologue of the catalytic subunit of human Protein Phosphatase 6 (PP6) [25–27]. We recently found that *PpV* is essential for *Drosophila* embryonic development and acts in parallel to *trbl* for timely cell cycle remodeling in interphase 14. Embryos lacking maternal PpV (embryos derived from *PpV* germline clones) are embryonic lethal before hatching. During blastoderm stage, about 30–50% of the embryos postpone cell cycle remodeling by one nuclear cycle [28].

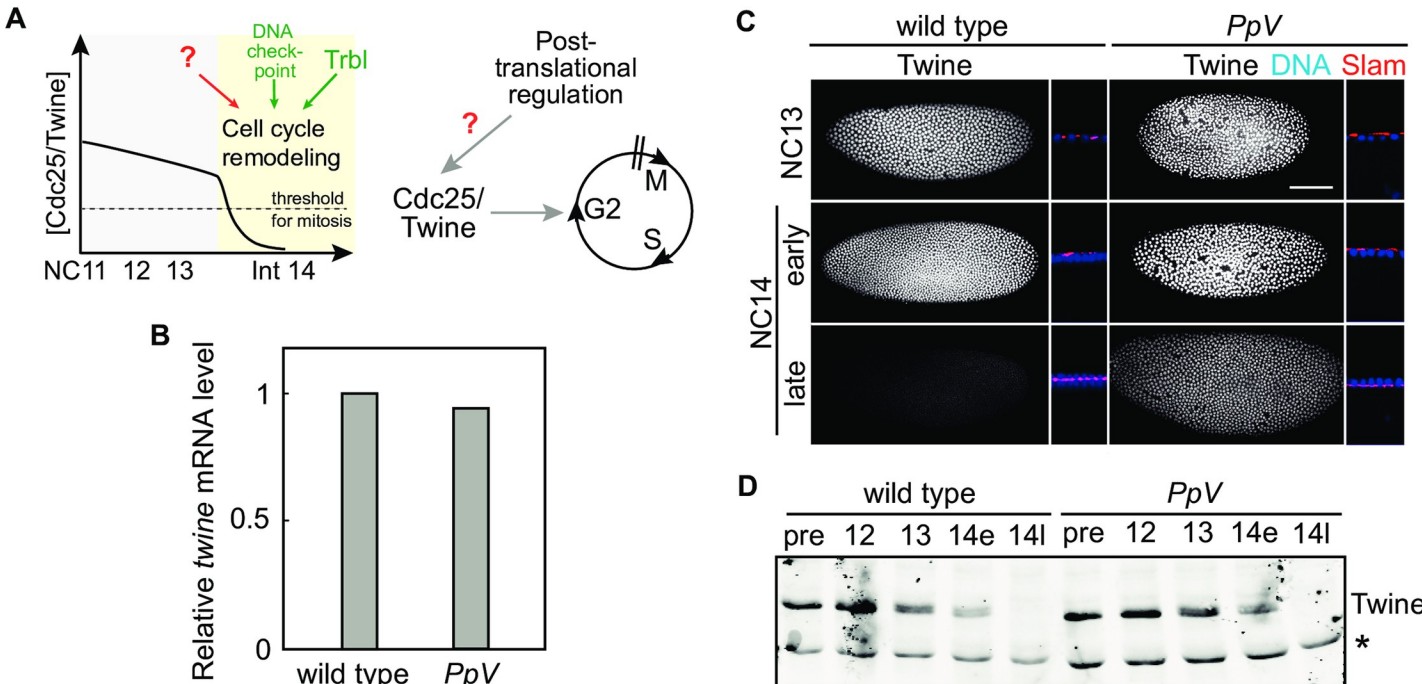

**Fig 1. Twine protein dynamics in *PpV* embryos. (A)** Schematic temporal profile of Cdc25/Twine during *Drosophila* early embryogenesis. The levels of Twine are post-translationally regulated by unknown mechanisms. **(B)** Quantification of *twine* mRNA levels by quantitative RT-PCR. Normalized by *GAPDH*. Bars indicate mean values (1:0.94, N = 2). **(C)** Fixed wild type embryos and *PpV* mutants stained for Twine (white), Slam (red), and DNA (DAPI, blue). Sagittal sections with Slam staining and nuclear morphology allow approximate staging of the embryos. Slam is restricted to the furrow canal marking the advance of furrow invagination during cellularization. NC, nuclear cycle. Scale bar 50 μm. **(D)** Western blot showing Twine expression profiles of nuclear cycle 12 to 14 in wild type and *PpV* embryos. Embryos were manually staged according to the nuclear density. Pre-blastoderm embryos (pre), nuclear cycle 12, 13, 14 early (14e) and late (14l). Cross-reacting band (marked by *) serves as the loading control.

Here, we hypothesize that *PpV* controls remodeling of the nuclear cycle through Cdc25/Twine. We investigate the functional and molecular relationship between *PpV* and *twine*. Given the phenotypic variability and the rapid change of Twine protein levels during early embryonic cleavage cycles, we applied fluorescence fluctuation analysis (FFA) as a non-invasive method to follow the absolute concentration of Twine-GFP in single living embryos from wild type, *PpV* and *trbl* mutants. We found that *PpV* is required for appropriately low Twine levels prior to *trbl*-mediated Twine destabilization and corresponding to cell cycle remodeling. We identified *PpV* dependent phosphosites in Twine by mass spectrometry. These phosphosites are relevant for controlling Cdc25/Twine protein dynamics and timing cell cycle remodeling as revealed in transgenic embryos.

## Results

### Persisting Cdc25/Twine protein levels in *PpV* mutants

The induced destabilization of Twine is achieved by a drop in the half life from about 20 min to 5 min in interphase 14, and *trbl* contributes to this destabilization [19, 21]. However, since only 10% of *trbl* mutants undergo an extra nuclear division, and *trbl* RNAi resulted only partial stabilization of Twine, other yet unknown factors may also play a role [19, 28]. Searching for additional factors, we found *PpV* to be involved in controlling Twine protein levels.

When investigating genetic interactions of *PpV* and *twine*, we found that *twine* is a mild dominant suppressor of *PpV*. We observed hatching larvae from *PpV* mutants that were

heterozygous for *twine* (>20 out of 1000 eggs), whereas no hatching larvae were observed from *PpV* mutants in wild type background. This interaction occurs on post-translational level as mRNA levels of *twine* are similar, but protein levels are higher in *PpV* mutants than in wild type. Quantitative PCR detected comparable levels of *twine* mRNA in wild type and *PpV* embryos (Fig 1B). Using immunostaining with morphological markers (nuclear and furrow lengths), we found that Twine appeared to persist longer in many but not all cellularizing *PpV* embryos (Fig 1C). Whereas western blot analysis of embryos staged according to early and late cellularization did not reveal an obvious difference (Fig 1D).

The difference between western blot analysis and immunostaining may be due to imprecise staging or embryo-to-embryo variation, which is obvious in *PpV* mutants. Moreover, western blot employed lysates derived from multiple embryos, thus revealing averages. To circumvent the problems of precise staging, phenotypic variability, and reliable quantification, a non-invasive quantitative method is needed to follow the expression levels of Twine during the course of nuclear cycles and cellularization in single selected embryos. Ideally, absolute concentrations are measured to allow a direct comparison between genotypes and make measurements independent of imaging and experimental conditions.

## Fluorescence fluctuation analysis (FFA) detects protein dynamics *in vivo*

We applied fluorescence fluctuation analysis (FFA), a simplified version of fluorescence correlation spectrometry (FCS), to trace the profiles of Twine-GFP protein in individual embryos with absolute time and concentration [29–32] (S1A Fig). Fluorescence fluctuations in a fixed focal volume depend on the fluorophore concentration, mobility, and volume size. From the fluctuation trace, the average number of molecules per focal volume can be determined. Knowing the focal volume, the absolute concentration can be calculated [33, 34]. For robust numbers, we averaged over multiple 10 sec fluctuation traces (S1E and S1F Fig), yielding a minute-scale temporal resolution. We experimentally determined a focal volume of 0.4±0.1 fl by measuring the xyz point spread functions of 100 nm microspheres in pre-blastoderm embryos [35] (S1B–S1D Fig).

We validated the FFA method with embryos containing one or two copies of nuclear GFP (nlsGFP) transgene. The copy number of nlsGFP transgene approximately corresponds to the protein content as confirmed by an about twofold increase in western blot (S1G Fig, S1 Data). Compared to this, FFA revealed a reliable twofold increase of the nuclear concentration, from 424±72 nM to 887±203 nM in embryos with one and two copies of the nlsGFP (S1H Fig, S1 Data). These data indicate that FFA can be applied in *Drosophila* embryos to non-invasively quantify absolute protein concentrations.

Next, we applied FFA to detect Twine protein levels. We employed a genomic Twine-GFP transgene, which functionally complements *twine* mutants (Fig 2A). We recorded fluorescence traces in embryos with one copy of Twine-GFP on top of two endogenous copies of *twine* at interphase 14. Having measured the relation of fluorescence and concentration in one pixel, we derived a spatial concentration map for the same image (Fig 2B). In this case, measurements revealed a concentration of 98.5 nM in the focal volume, and averaging over the field of view resulted in a nuclear concentration of 101 nM.

We then determined the temporal concentration profile by repetitive measurements in interphases of the nuclear cycles and during cellularization/interphase 14. In a selected embryo, we measured the nuclear concentration of Twine-GFP starting with interphase 11 until late cellularization. In particular, during cellularization, we measured the concentration every few minutes to better follow the decay dynamics. We found that the nuclear concentration dropped from more than 300 nM in interphase 11, to about 150 nM at the onset of

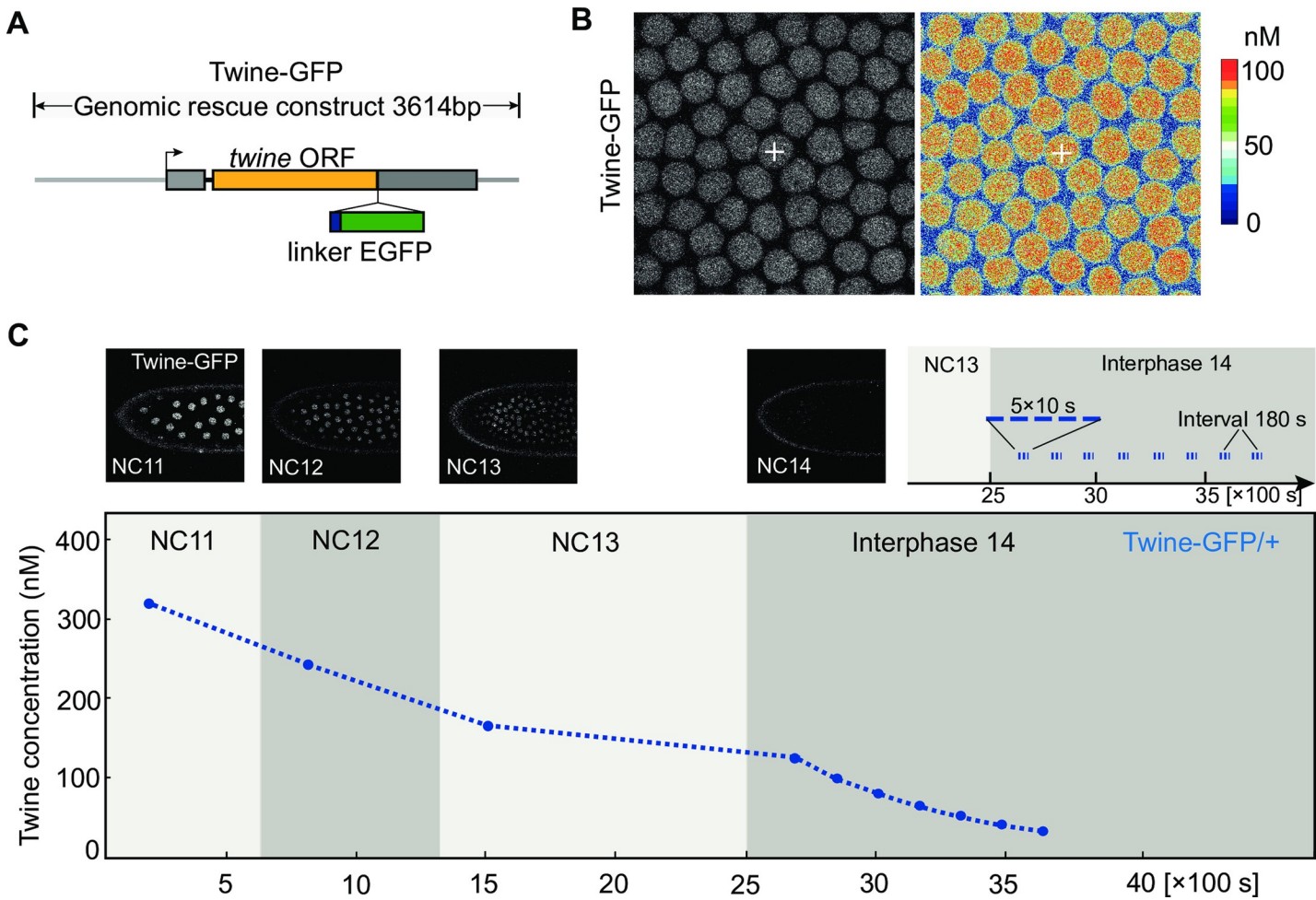

**Fig 2. Concentration profile of Twine-GFP by fluorescence fluctuation analysis.** (**A**) Scheme of Twine-GFP genomic rescue construct. *Twine* open reading frame (ORF) is indicated in orange. Linker sequence (Stuffer) is indicated in blue box and EGFP is in green box. Untranslated regions (UTR) are indicated in grey boxes. 3,614 bp genomic region is depicted in grey line. (**B**) Spatial concentration map of Twine-GFP in cortical nuclei of a wild type embryo with one copy of Twine-GFP transgene at interphase 14/early cellularization. Heat-map showing the scale of calculated absolute concentration of Twine-GFP. White crosses indicate the position of measurement/focal volume. (**C**) Experimental scheme of temporal Twine profiles. Upper panel: Images from a recording and the recorded time points in interphase 14. Lower panel: Measured Twine-GFP concentration in a wild type embryo with one copy of Twine-GFP transgene. S, second. NC, nuclear cycle.

cellularization (Fig 2C). Afterwards, Twine-GFP rapidly dropped below the detection levels within 20 to 30 min in interphase 14. Our data are consistent with the published relative profiles of Twine-GFP [21].

## Dynamics of Twine protein depends on *PpV* and *tribbles*

We tested the hypothesis that *PpV* is involved in controlling Twine protein levels by measuring Twine-GFP profiles in wild type, *PpV* and *trbl* mutant embryos using FFA. Although Twine-GFP levels were initially within the same range during nuclear cycles, we detected significantly higher levels in *PpV* embryos at early interphase 14. Here, Twine-GFP levels were in average 1.7-fold higher in *PpV* mutants than in wild type and *trbl* embryos (Fig 3A).

We analyzed the Twine profiles in a greater temporal resolution during cellularization/ interphase 14, when Twine swiftly decays. To document embryo-to-embryo variations and for ease of comparison between the genotypes, we fitted an exponential function to the individual

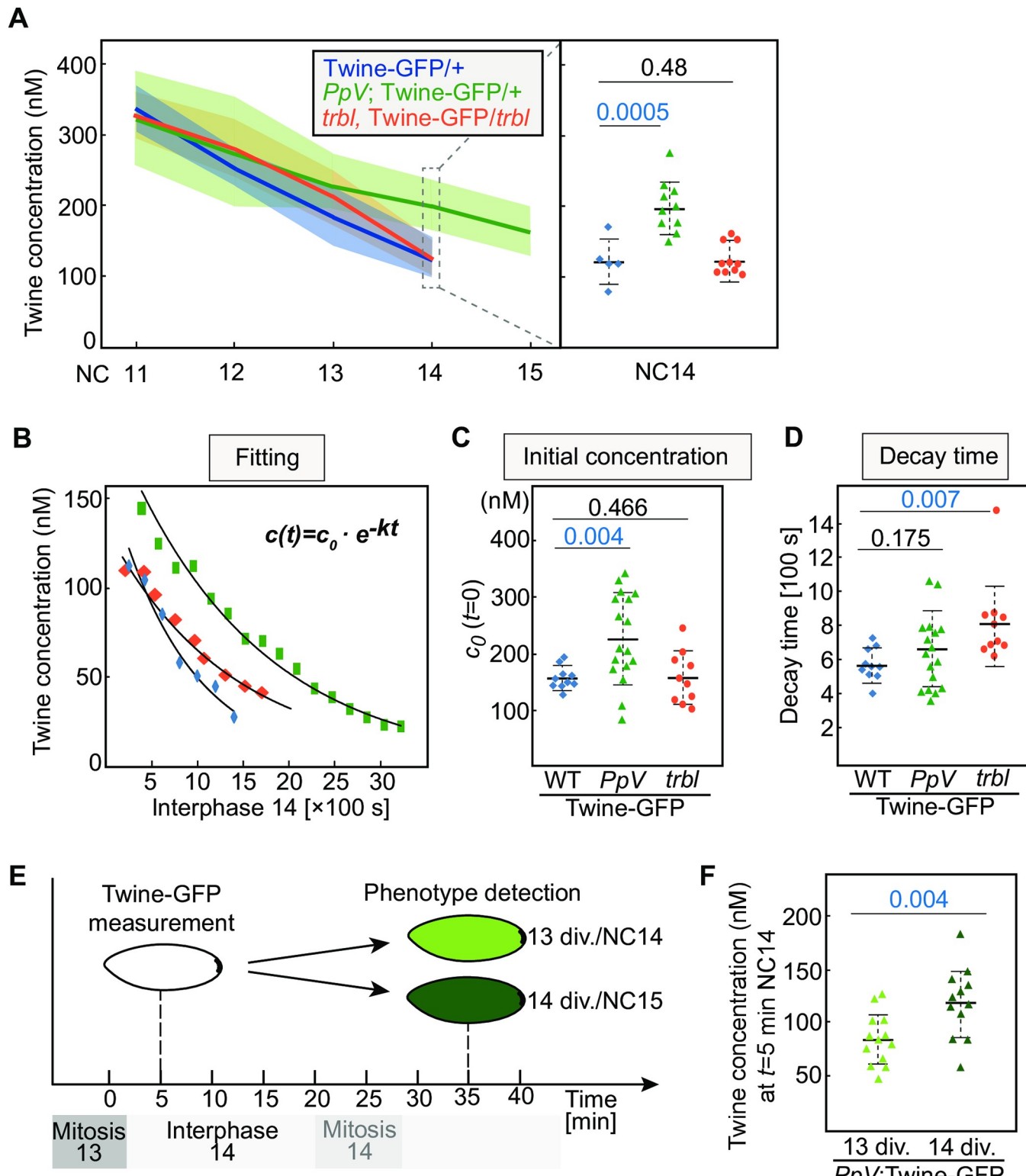

**Fig 3. Dynamics of Twine protein depends on *PpV* and *tribbles*. (A)** Absolute concentration of Twine-GFP during syncytial cycles. Genotypes are indicated. Embryos are from wild type, *PpV* germline clones, and maternal and zygotic homozygous *trbl* mutants, all of which are with one copy of Twine-GFP. Standard deviations of the mean value are indicated by faint colors. Right column showing distributions of individual embryos at cycle 14. NC, nuclear cycle. **(B)** Representative time courses for individual wild type (blue), *PpV* mutant (green), and *trbl* mutant (red). Exponential fitting provides two parameters for each embryo: initial concentration ($c_0$) and the decay time. S, second. **(C, D)** Distribution of calculated initial concentration at $t = 0$ of interphase 14 and the decay

time for wild type, *PpV* mutant, and *trbl* mutant embryos. **(E)** Experimental scheme of the phenotype detection of *PpV*; Twine-GFP/+ embryos. The measurements were carried out 5 min after mitosis 13. 30 min after the measurements, the embryos were observed under widefield microscopy and the extra division phenotype was detected. **(F)** Distribution of calculated average protein concentration at $t = 5$ min for *PpV* embryos with 13 and 14 nuclear divisions. Mean, bold line. Standard deviation, dashed line. The numbers indicate the statistical significance (p value by Student's t-test) for the difference between two distributions. Source data are shown in S1 Data and S2 Fig.

traces which yielded two parameters (Fig 3B). The initial concentration ($c_o$) is the Twine levels at the onset of interphase 14 ($t = 0$). The exponential constant ($k$) provides a phenological decay time, which represents the difference between translation and degradation rates. While half life refers to as the molecular lifetime of Twine protein molecules, the phenological decay time is based on a measurement of steady state concentration reflecting the bulk of Twine molecules. With the assumption of a constant translation rate, the changes in phenological decay time indicate changes in protein stability. To assess the phenotypic variability, we measured the profiles in 10 individual embryos for each genotype, at least.

We obtained a narrow distribution of initial concentrations of 160±21 nM and a decay time of less than 10 min in wild type embryos (Fig 3C and 3D, S1 Data). The distributions of these two parameters were much wider in the mutants. Especially, the initial concentration in *PpV* embryos appeared as an almost bimodal distribution ranging from 100 nM to more than 300 nM. The average initial concentration of 236±81 nM was significantly higher than the value in wild type and *trbl* embryos. Conversely, the average decay time was not significantly changed in *PpV* mutants. Consistent with previous reports of Trbl induced Twine degradation [19], the decay time was significantly increased to 13.5±4.27 min in *trbl* mutants as compared to wild type (Fig 3C and 3D). These data indicate that *PpV* and *trbl* control the protein amount of Twine in two different manners. *PpV* ensures appropriately low levels of Twine prior to interphase 14, whereas *trbl* contributes to the induced Twine degradation in response to zygotic genome activation during cellularization.

*PpV* mutants are characterized by an extra nuclear cycle in about 30–50% of the embryos [28]. Given the bimodal distribution of initial Twine concentration, we hypothesized that *PpV* mutants with increased initial Twine levels were more likely to undergo an extra division than that *PpV* mutants with normal levels of Twine. To establish the correlation between initial Twine levels and the cell cycle phenotype, we measured Twine-GFP levels 5 min after mitosis 13 and scored the embryonic cell cycle later on (Fig 3E). In both populations of *PpV* embryos (13 versus 14 divisions), we observed wide and overlapping distributions of Twine-GFP (Fig 3F, S1 Data). Importantly, the averaged Twine concentration was significantly higher in embryos with an extra division. These data indicate that the embryos with higher Twine levels are more likely to undergo an extra division, consistent with the function of Twine as an activator of Cdk1. However, the overlapping distributions also suggest that other, yet unknown, factors besides Twine contribute to the control of mitotic entry. Taken together, our measurements indicate that *PpV* controls Twine protein levels and ensures a low embryo-to-embryo variability prior to interphase 14.

## Twine is hyperphosphorylated in *PpV* mutants

As a first step into the molecular mechanism, we analyzed the phosphorylation state of Twine in early *Drosophila* embryos. We isolated Twine-GFP with a single chain GFP nanobody from wild type and *PpV* mutant embryos (Fig 4A), and subjected the isolated protein to phosphopeptide analysis by mass spectrometry. We achieved about 70% coverage and detected several phosphopeptides (Fig 4B, S1 Table). Two paired phosphosites were identified in both wild type and *PpV* embryos: Thr203+Ser205 and Thr394+Ser396 (Fig 4B, S2 Table), indicating

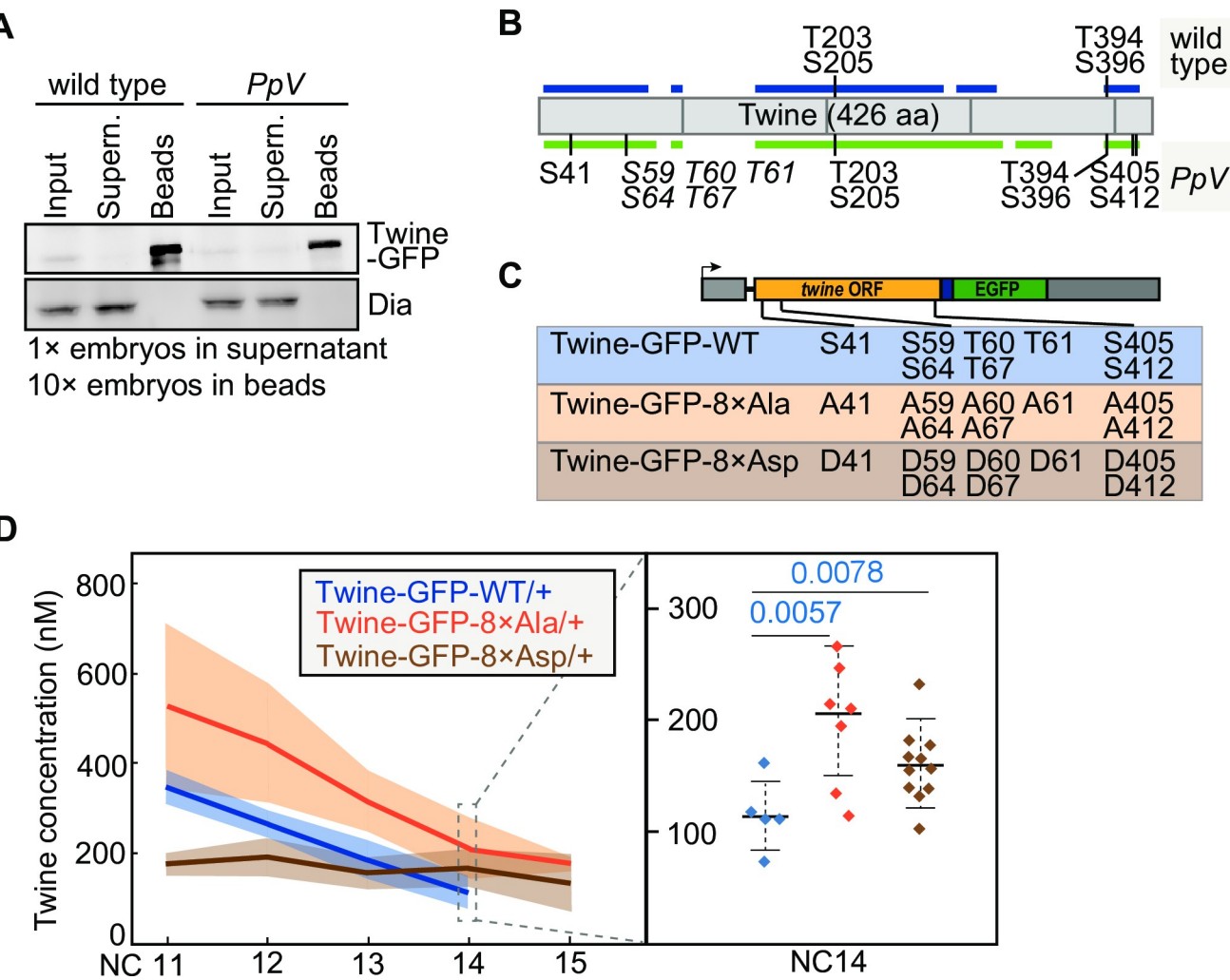

**Fig 4. Identification of *PpV* dependent phosphorylation sites in Twine.** (A) Twine-GFP was isolated with GFP-binder bound to beads from extracts of timed (0–1.5 h) wild type or *PpV* embryos with Twine-GFP transgene. Western blot with GFP and control (Dia) antibodies. Dia is a ubiquitously expressed formin. (B) Schematic diagram showing the position of Twine peptides covered in the analysis and the identified phosphosites. Italic types indicate ambiguous phosphosites. Coverages in wild type and *PpV* mutants are shown in blue (59.4%) and green (70%), respectively. Aa, amino acid. (C) Schematic diagram showing the transgenic Twine-GFP construct with non-phosphorylatable and phosphomimetic mutations. Eight *PpV* dependent residues are shown in the box of Twine-GFP-WT. Non-phosphorylatable mutant Twine-GFP-8×Ala indicates the replacement of the eight phosphorylation residues to alanine/A. Phosphomimetic mutant Twine-GFP-8×Asp indicates the replacement of the eight phosphorylation residues to aspartic acid/D. (D) Absolute concentration of Twine-GFP during syncytial cycles. Genotypes are indicated. The data of wild type control are the same as which in Fig 3A. Standard deviations of the mean value are indicated by faint colors. Right column showing distributions of individual embryos at cycle 14. NC, nuclear cycle. Mean, bold line. Standard deviation, dashed line. The numbers indicate the statistical significance (p value by Student's t-test) for the difference between two distributions. Source data are shown in S1 Table, S2 Table and S1 Data.

constitutive phosphorylation which does not depend on *PpV*. In addition, we identified eight sites specifically in *PpV* mutants, indicating that their dephosphorylation depends on *PpV*. These eight *PpV* dependent phosphosites clustered in three regions. The experimental data are strong and unambiguous for Ser41 and a paired site close to the C-terminus, Ser405 and Ser412. The other five sites clustered in a region between 59 and 67. The phosphosite mapping result indicates that at least one among the five residues in that region are hyperphosphorylated in *PpV* mutants (Fig 4B, S2 Table). Thus, our data indicate that *PpV* is required for dephosphorylation of Twine at three positions, at least. Our experimental approach does not

allow to distinguish, however, whether *PpV* acts directly on Twine or indirectly via additional and unknown protein kinases or phosphatases.

Having revealed Twine hyperphosphorylation in *PpV* mutants, we further tested the functional relevance of these sites on Twine protein dynamics and the timing of cell cycle remodeling. We mutated the amino acid residues of all eight *PpV* dependent phosphosites in Twine. We generated Twine-GFP genomic transgenes, in which the respective serine/threonine residues were mutated to either aspartic acid (Twine-GFP-8×Asp, phosphomimetic) or alanine (Twine-GFP-8×Ala, non-phosphorylatable) (Figs 2A and 4C). We crossed the transgenes into *twine* mutant background and tested for complementation. One copy of both transgenes complemented the *twine* female sterility. When making the transgenes homozygous (two copies of the transgenes), we observed a dose dependent lethality of the phosphomimetic Twine-GFP-8×Asp but not by the non-phosphorylatable or wild type transgene. Such a neomorphic phenotype indicates that the Twine-GFP-8×Asp protein has acquired novel activities such as novel substrates, making interpretation of the phenotypes difficult.

We quantified Twine protein levels by FFA in embryos expressing one copy of mutated Twine-GFP in wild type background. We detected strongly increased levels of the non-phosphorylatable Twine-GFP-8×Ala protein already during nuclear cycle 11. In contrast, levels of the phosphomimetic form were strongly reduced but remained constant during nuclear cycles (Fig 4D, S1 Data). At cycle 14, when wild type Twine-GFP levels have dropped to about 119 ±28 nM, both mutant protein forms were significantly higher with 206±58 nM for the 8×Ala and 169±36 nM for the 8×Asp Twine-GFP transgenes, respectively. Consistent with the higher initial concentrations of Twine-GFP at the onset of interphase 14, we observed an extra mitosis in a small fraction of the embryos for both phosphomimetic (7%, N = 14) and non-phosphorylatable Twine-GFP (17%, N = 23).

In summary, we detected *PpV* dependent phosphosites in Twine, which are relevant for protein stability and for Twine's function in mitotic control. Mutation of these phosphosites leads to changes of protein levels and corresponding to an extra nuclear division prior to cellularization, albeit at low penetrance. We conclude that these phosphosites are important and physiologically relevant for ensuring the proper developmental profile of Cdc25/Twine protein levels, and thus proper developmental control of nuclear cycles.

## Discussion

In this study, we provide a versatile quantitative and non-invasive method, fluorescence fluctuation analysis (FFA), for measuring absolute protein concentration with a high spatiotemporal resolution. We applied FFA to detect the absolute concentration of tagged Cdc25/Twine in living embryos, and correlated the Twine concentration to the developmental path. Because this approach is based on frequency but not amplitude of the fluorescent signal [29, 31], experimental conditions such as changing laser power or photo-bleaching do not obscure the measurements.

Cdc25 activity is determined by protein amount as well as post-translational modifications such as phosphorylation [24, 36–39]. However, it is yet unclear how the activity and amount of Cdc25/Twine are regulated in early *Drosophila* embryos. As Twine protein and mRNA are maternally provided and *twine* RNAi does not alter cell cycle remodeling [19], its regulation in early embryos is achieved post-translationally. Firstly, Twine protein levels are set low at the onset to allow a swift degradation within the first 20 min of cellularization/interphase 14. Secondly, the half life of Twine strongly decreases in interphase 14 in response to the onset of zygotic gene expression. The molecular mechanisms for both modes have remained unclear. Analysis has been hampered by the lack of a simple and non-invasive assay to measure the

changing of Twine protein levels and correlate to the consequent phenotype within selected embryos. Previous studies estimated that a threshold concentration of about 45 nM for the second Cdc25 in *Drosophila*, String, during the entry into mitosis in gastrulation stage [40]. Our FFA measurements revealed a Twine nuclear concentration of 41±9 nM at 20 min into interphase 14, when the decision for an extra mitosis is made (S1 Data).

We have previously shown that the *Drosophila* homologue of PP6, *PpV*, acts in parallel with *trbl* in remodeling of the nuclear cycle [28]. Here, we reveal a mechanism that *PpV* controls the protein levels of Twine through modulating its phosphosites, thus controlling the timing of cell cycle remodeling. We propose that *PpV* controls Twine during nuclear cycles resulting in appropriately and uniformly low protein levels prior to interphase 14. In contrast, *trbl* and other unknown factors are involved in the swift destabilization of Twine in interphase 14. Importantly, we show that *PpV* keeps a low embryo-to-embryo variation of Twine protein. It is a remarkable property of *Drosophila* embryos that no variation is observed for the number of nuclear cycles and thus the timing of cell cycle remodeling in wild type. Safeguarding mechanisms for phenotypic uniformity are crucial, especially when thresholds and gradients (temporal decay of Twine) are centrally involved in timing.

Notably, albeit clearly correlating, Twine protein levels may not be the exclusive parameter of cell cycle remodeling, since not all embryos with high Twine concentration underwent an extra mitosis. Likewise, a tripling of *twine* gene dose to 6-fold only rarely leads to an additional nuclear cycle [18]. In a simple model based on Twine protein destabilization in interphase 14, a dose-dependent phenotype would be expected for the cell cycle remodeling. Other factors may also contribute to this process, such as Wee1. Embryos lacking maternal Wee1 show a cell cycle arrest prior to the remodeling. Wee1/Myt1 and Cdc25 control Cdk1 in pair through T14Y15 phosphosites, and both are regulated by checkpoint kinase Chk1 [41, 42].

Biochemical analysis clearly revealed that the phosphorylation status of Twine in *PpV* embryos is different than in wild type. It is clear that such an analysis does not reveal whether *PpV* acts directly or indirectly on the identified phosphosites of Twine. A comprehensive biochemical and enzymatic analysis with purified components would be required to define the molecular relationship. In our study, we focus on the functional relationships. We measured Twine stability by FFA in transgenic embryos, in which all of the eight *PpV* dependent serine/threonine residues were substituted with either alanine or aspartic acid. The mutations did not affect the general structure and the function of Twine, since both forms can substitute for wild type Twine, *i.e.*, complemented the *twine* mutation. Despite this, the non-phosphorylatable mutant showed strongly increased protein levels already in nuclear cycle 11, whereas the levels of the phosphomimetic form were low but remained very stable during syncytial cycles. This and the fact that the phosphomimetic form shows a dose-dependent neomorphic activity indicate that the Twine phosphosite mutants are different than the Twine in *PpV* mutants. This difference may be due to contributions of other protein kinases and phosphatases in regulation of Twine at these sites. The difference may also be due to the degree of phosphorylation. In the phosphosite mutants, the amino acid residues in every Twine molecule were changed, whereas only a fraction of them may be phosphorylated in *PpV* mutants. Other studies about the phosphorylation status of Twine showed that replacing eight phosphosites to alanine within the Destruction Boxes of Twine led to low Twine levels in nuclear cycle 12, and replacing the three Cdk1-mediated phosphosites to alanine resulted prolonged nuclear cycles 10–13 [21, 43]. We do not know how Twine activity is altered in the phosphosite mutants. The mutated residues may affect kinase activity, substrate specificity, or interactions with other proteins. It is also conceivable that Twine undergoes a reaction cycle of successive and transient phosphorylation and dephosphorylation.

Taken together, we reveal a novel mechanism for Cdc25/Twine function in *Drosophila* embryonic cell cycle remodeling. We propose that the mitotic decision is determined by two parameters: (1) *PpV* controlled Twine levels at the onset of interphase 14, and (2) the speed of induced Twine degradation during interphase 14, which involves *trbl* and other factors. Future experiments will define the mechanisms how *PpV* controls Twine protein levels and how multiple redundant pathways contribute to the robustness of cell cycle remodeling.

## Materials and methods

### Genetics

Fly stocks were obtained from the Bloomington Drosophila Stock Center [44], if not otherwise noted. Genetic markers and annotations are described in Flybase [45]. Following fly strains and mutations were used: *trbl*[EP1119], *twine*[HB5], and *PpV*[X9] [28]. Following transgenes were used: Twine-GFP (genomic transgene integrated at the landing site attP2/68A4; S. Blythe, S. Di Talia, and E. Wieschaus), Twine-GFP-8×Ala, Twine-GFP-8×Asp, and nlsGFP Frt[18E]. For *PpV* mutants, virgin females *PpV*[X9] Frt[18E] hs-Flp[122] / FM7 were crossed with *ovo* [D] Frt[18E] / Y males. Germline clones of *PpV* were induced with the Flippase/Frt system by heat-shock (2× 1 h in the first and second instar larvae).

### Molecular genetics, cloning, constructs

The Twine-EGFP transgene was synthesized as a 3.6 kb complementing genomic fragment [16] with an in-frame insertion of a *Drosophila* codon optimized EGFP at the C-terminus including a linker sequence (S. Blythe, S. Di Talia, and E. Wieschaus). The constructs of mutated Twine phosphosites were synthesized by Eurofins Genomics and cloned into the KpnI/BamHI fragment of Twine-EGFP-pBABR plasmid.

### RNA isolation, quantitative RT-PCR

Quantitative PCR and data analysis were carried out according to the protocols of SYBR Green Real-Time PCR Master Mixes and qPCR system software (Thermo Fisher Scientific). cDNA template was synthesized by reverse transcription of *Drosophila* total RNA, which was isolated by using TRIzol total RNA isolation protocol (Invitrogen). The following primer pairs were used: *twine* qPCR primers BL16 (GAGTTCCTTGGCGGACACAT) and BL17 (CAGGA TAGTCCAGTGCCGGAT); *GAPDH* qPCR primers MP37F (CACCAGTTCATTCCCAAC TT) and MP37R (CTTGCCTTCAGGTGACGC) [46].

### Antibodies

Following antibodies were used: Dia (rabbit, guinea pig) [47], Slam (rabbit, guinea pig) [48], Twine (rat) [21], and GFP (mouse; Chemicon, CA, USA).

### Embryo microinjection

Embryos were dechorionated with 2.5% sodium hypochlorite bleached for 90 s, dried in a desiccation chamber for 10 to 12 min, then covered by halocarbon oil. Glass capillaries with internal filament were pulled as needles. For focal volume detection, 100 nm FluoSpheres Fluorescent Microspheres (Molecular Probes, OR, USA) were diluted to 1:10,000 and injected posteriorly to the embryos. For transgenesis, DNA was injected at 0.1 μg/μl prior to the pole cell formation [49].

## Fluorescence fluctuation analysis

Following female genotypes were utilized: Wild type: Twine-GFP/+. *PpV*: *PpV*[X9] /*ovoD*; Twine-GFP/+. *Trbl*: *trbl*, Twine-GFP/*trbl*. Phosphosite mutants: Twine-GFP-8×Ala/TM3, Twine-GFP-8×Asp/TM3. Fluctuation traces were recorded with a 63× oil immersion objective (Planapochromat, NA1.4/oil) and a GaAsP detector on a confocal microscope (Zeiss LSM780) in fluorescence correlation spectrometry (FCS) mode. The data were analyzed as previously reported [33, 34]. Briefly, the intensity traces were analyzed using the number and brightness analysis. Usually 5 traces of each 10 s length were used for calculation of one data point. Time average $<i>$ and variance $\sigma^2$ were computed for each trace. From these values, we obtained the average brightness per molecule as $<e> = (\sigma^2 / <i>) - 1$, the average number of molecules within the focal volume as $<n> = <i> / <e>$, and the absolute concentration as $<c> = <n>$ / (focal volume × Avogadro constant). Determination of the focal volume was conducted using an approach previously described [35]. Images of the spatial concentration map were processed with ImageJ/FIJI. For the measurements of Twine-GFP during nuclear cycles 11–14 (15), we started the recording in early interphases of each cycle when a stable nuclear Twine-GFP signal appeared and a stable nuclear position was observed. For the time dependent decay of Twine-GFP during interphase 14, at least five data points were used for fitting in an exponential curve by linear regression analysis. Fitting curve of exponential trend line was represented by formula $c(t) = c_0 \cdot e^{-kt}$, and coefficient of determination by value $R^2$, which were in the range of $R^2 > 0.97$. The initial concentration ($c_0$) was defined as the Twine-GFP levels at the onset of interphase 14 ($t = 0$) when a complete loss of nuclear Twine-GFP signal in mitosis 13 was observed and the signal started to appear again in the following interphase 14. The decay time was calculated from the exponential constant ($k$) by formula–(ln 0.5) / $k$. The statistical significance (p value) of differences between the measured distributions was calculated by Student's t-test. Details of the analysis and the MATLAB code are available upon request.

## Identification of phosphosites in Twine by mass spectrometry

Dechorionated syncytial (0–1.5 h) Twine-GFP/Twine-GFP embryos and embryos from *PpV* germline clones with Twine-GFP were collected in large batches and snap frozen in liquid nitrogen. Each 10,000 embryos were lysed with a Dounce homogenizer in 1 ml lysis and washing buffer (50 mM Tris/HCl [pH 7.4], 500 mM NaCl, 1 mM DTT, 0.5 mM EDTA, 0.5% Tween 20, 1% Phosphatase Inhibitor Cocktail 3 (Sigma-Aldrich), 1 Tablet/50 ml Protease Inhibitor Cocktail (Roche)). The preparatory experiment was in the scale of about 50,000 embryos. The embryonic lysate was centrifuged twice at 14,000 rpm at 4°C for 15 min, and the supernatant was transferred into a new Eppendorf tube. GFP-TrapA agarose beads (20 μl; Chromotek) were washed thrice with lysis and washing buffer and added to 1 ml of cleared embryonic lysate. After rotating on a wheel for 1 hour at 4°C, spinning at 800 rpm for 2 min, and washing Twine-GFP thrice with lysis and washing buffer, protein was eluted from the beads in Laemmli buffer. Samples were run on SDS-PAGE gradient 4–12% MOPS buffered system and bands were excised and processed. Band slices were digested with trypsin overnight at 30°C and the peptide extracted the next day. Samples were resuspended in 1% formic acid and a 15 μl aliquots of each sample were run either before or after phosphopeptide enrichment. Phosphopeptides were enriched using Ti 4+ IMAC (ReSyn Biosciences) on an UltiMate 3000 RSLC nano system (Thermo Scientific) coupled to an LTQ OrbiTrap Velos Pro (Thermo Scientific). Peptides were initially trapped on an Acclaim PepMap 100 (C18, 100 μm × 2 cm) and then separated on an EasySpray PepMap RSLC C18 column (75 μm × 50 cm) (Thermo Scientific) over a 120 min linear gradient. The data was analyzed by Proteome Discoverer 1.4 (Thermo Scientific) using Mascot 2.4 (Matrix Science) as the search engine. The detailed data sets are available upon request.

## Supporting information

**S1 Fig. Fluorescence fluctuation analysis for absolute protein concentration profile. (A)** Scheme of fluorescence fluctuation analysis application in living embryo. **(B)** Principle of fluorescence fluctuation analysis. Passing through the focal volume leads to changes in the fluorescence signal. The frequency of these changes depends on the concentration. **(C)** Isosurface representations of a microsphere. **(D)** The volume of the point spread function. **(E)** Fluctuation trace of a measurement (10 seconds). **(F)** Time average and variance are computed from the fluctuations trace, which allows calculation of average molecular brightness and average number of molecules within the focal volume, as well as absolute protein concentration. **(G)** Western blot with extracts of the embryos (0–3 h) with 0×, 1×, 2× copies of nlsGFP transgene. Loading control by Dia protein, which is a ubiquitously expressed formin. Quantification by densitometry (N = 3, biological replicates on the same blot, average with standard deviation). **(H)** Concentration per focal volume as determined by fluorescence fluctuation analysis of GFP fluorescence. Mean, bold line. Standard deviation, dashed line. Source data are listed in S1 Data.
(PDF)

**S2 Fig. Source data of Fig 3.** Exponential fitting of fluorescence fluctuation analysis measurements. Formulas representing fitting curve of exponential trend line $c(t) = c_0 \cdot e^{-kt}$, and coefficient of determination value $R^2$. Unit of the x-axis is second.
(PDF)

**S1 Table. Comparison of non-phosphopeptides in MS analysis of wild type and *PpV* embryos.** The average peak area of five peptides were determined by XIC (extracted ion chromatograms) and compared between the wild type (wt) and the *PpV* mutant samples. The AA ratio of wt/*PpV* averages at 1.78 indicating that non-phosphopeptides were consistently about 1.8-fold more abundant in the wild type compared to *PpV* mutant samples.
(DOCX)

**S2 Table. Analysis of phosphorylation sites by mass spectrometry.** Results from Mascot data analysis of mass spectrometry (ms/ms) spectra. Peptide sequences are depicted of the Cdc25/Twine as predict from fragmentation spectra. Predicted phosphorylation sites are marked in red (bold colors for unambiguous annotations). Although peptides were identified independently in many cases only highest scoring peptides are included. Observed mass/charge (M/Z) values indicate the result of the measurement and the calculated relative molecular weight (Mr) from the M/Z is indicated as experimental (expt) Mr in Dalton (Da). Mr calc depicts the calculated relative molecular weight in Dalton (Da) as calculated from the expected Mr from the database. Ppm indicates the error value between Mr expt and Mr calc and the ion score indicates the number of spectral ions matching the annotated fragments in the database. The expectation value is a statistical representation of the ion score expressed as p value (Student's t-test).
(DOCX)

**S1 Data. Source data in Excel sheets with the data as shown in the Figs 3, 4 and S1.**
(XLSX)

## Acknowledgments

We are grateful to S. Blythe, S. Di Talia, C. Lehner, S. Luschnig, and E. Wieschaus for materials or discussions. We thank R. Webster, D. Lamont, and K. Beattie of the School of Life Sciences, University of Dundee, UK, for sample processing and mass spectrometry. We acknowledge

service support from the Developmental Studies Hybridoma Bank created by NICHD of the NIH/USA and maintained by the University of Iowa, the Bloomington Drosophila Stock Center (supported by NIH P40OD018537), the FlyBase (supported by NIH U41HG000739), and the Drosophila Genomics and Genetic Resources, Kyoto.

## Author Contributions

**Conceptualization:** Jörg Großhans.

**Data curation:** Boyang Liu, Ingo Gregor, H.-Arno Müller.

**Formal analysis:** Boyang Liu, Ingo Gregor, H.-Arno Müller.

**Funding acquisition:** Jörg Großhans.

**Investigation:** Boyang Liu, Jörg Großhans.

**Methodology:** Boyang Liu, Ingo Gregor, H.-Arno Müller, Jörg Großhans.

**Project administration:** Jörg Großhans.

**Resources:** Jörg Großhans.

**Supervision:** Jörg Großhans.

**Validation:** Boyang Liu, Jörg Großhans.

**Visualization:** Boyang Liu, Jörg Großhans.

**Writing – original draft:** Boyang Liu, Jörg Großhans.

**Writing – review & editing:** Boyang Liu, Jörg Großhans.

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
