## [Decision Letter · Decision Letter 0]

13 Jan 2020

Dear Dr Grosshans,

Thank you very much for submitting your Research Article entitled 'Fluorescence fluctuation analysis (FFA) reveals PpV dependent Cdc25 protein dynamics in living embryos' to PLOS Genetics. Your manuscript was fully evaluated at the editorial level and by independent peer reviewers. The reviewers appreciated the attention to an important problem, but raised some substantial concerns about the current manuscript. Based on the reviews, we will not be able to accept this version of the manuscript, but we would be willing to review a revised version, which should include additional experimental work. We cannot, of course, promise publication at that time.

If you decide to revise the manuscript for further consideration at PLOS Genetics, please aim to resubmit within the next 60 days, unless it will take extra time to address the concerns of the reviewers, in which case we would appreciate an expected resubmission date by email to plosgenetics@plos.org.

[LINK]

We are sorry that we cannot be more positive about your manuscript at this stage. Please do not hesitate to contact us if you have any concerns or questions.

Yours sincerely,

Jean-René Huynh

Associate Editor

PLOS Genetics

Gregory P. Copenhaver

Editor-in-Chief

PLOS Genetics

Reviewer's Responses to Questions

**Comments to the Authors:**

Reviewer #1: The reviews were uploaded in a separate file.

Reviewer #2: In this manuscript, Liu et al address the mechanisms of cell cycle regulation at the maternal-to-zygotic transition. They focus on the regulation of Cdc25Twine levels, which were previously shown to be essential for cell cycle remodeling. This paper extends our understanding of Cdc25Twine regulation by showing that the levels of protein loaded in the embryo by the mothers increase when Protein Phosphatase V is deleted. As a consequence, a fraction of the embryos would not pause the cell cycle after 13 divisions, but would undergo a 14th mitosis prior to arresting, a typical phenotype observed when the regulation of the maternal-to-zygotic transition is perturbed. These are interesting insights. The authors also show that Twine is hyperphosphoryalted when PPV is deleted. While there are many open questions about the mechanistic action of those phosphorylation the fact that the effects of PPV are through Cdc25Twine phosphorylation (either direct or indirect) will be helpful for future studies. Overall, while many open questions remain on how PPV controls Cdc25Twine during early embryogenesis, collectively the insights presented in this paper are interesting and provide an advancement. I only have one comment. The authors should quantify the amount of cdc25twine maternal mRNA and compare it to wild type level. It is important to know whether the higher protein levels are also linked to higher mRNA levels. I might have missed this important control- in such case the authors need to discuss it more clearly.

Reviewer #3: See attachment.

**Have all data underlying the figures and results presented in the manuscript been provided?**

Reviewer #1: Yes

Reviewer #2: Yes

Reviewer #3: Yes

PLOS authors have the option to publish the peer review history of their article (what does this mean?). If published, this will include your full peer review and any attached files.

Reviewer #1: No

Reviewer #2: No

Reviewer #3: No

---

## [Decision Letter · Decision Letter 1]

25 Mar 2020

Dear Dr Liu,

We are pleased to inform you that your manuscript entitled "Fluorescence fluctuation analysis reveals PpV dependent Cdc25 protein dynamics in living embryos" has been editorially accepted for publication in PLOS Genetics. Congratulations!

Yours sincerely,

Jean-René Huynh

Associate Editor

PLOS Genetics

Gregory P. Copenhaver

Editor-in-Chief

PLOS Genetics

Comments from the reviewers (if applicable):

Reviewer's Responses to Questions

**Comments to the Authors:**

Reviewer #1: The authors answered my question about the phosphorylation sites identified in their first version of the manuscript quite clearly. Whereas it appeared, when I read the first version of the manuscript that 4 sites were dephosphorylated by PP5, it seems that a total of 8 phosphorylation sites are actually de-phosphorylated by this phosphatase. Therefore, the manuscript (with the de-phosphorylation defective variants) in its present form therefore is acceptable for publication in PlOS Genetics.

Reviewer #2: The authors have addressed my criticism

**Have all data underlying the figures and results presented in the manuscript been provided?**

Reviewer #1: Yes

Reviewer #2: Yes

PLOS authors have the option to publish the peer review history of their article (what does this mean?). If published, this will include your full peer review and any attached files.

Reviewer #1: No

Reviewer #2: No

**Data Deposition**

http://datadryad.org/submit?journalID=pgenetics&manu=PGENETICS-D-19-01957R1

**Press Queries**

---

## [Editor Report · Acceptance letter]

30 Mar 2020

PGENETICS-D-19-01957R1 

Fluorescence fluctuation analysis reveals PpV dependent Cdc25 protein dynamics in living embryos 

Dear Dr Liu, 

We are pleased to inform you that your manuscript entitled "Fluorescence fluctuation analysis reveals PpV dependent Cdc25 protein dynamics in living embryos" has been formally accepted for publication in PLOS Genetics! Your manuscript is now with our production department and you will be notified of the publication date in due course.

With kind regards,

Matt Lyles

PLOS Genetics

On behalf of:
